# Changes in the Gut Microbiota Composition of Juvenile Olive Flounder (*Paralichthys olivaceus*) Caused by Pathogenic Bacterial Infection

Dong-Gyun Kim [1],[†] , Su-Jeong Lee [2],[†], Jong Min Lee [3] , Eun-Woo Lee [2] and Won Je Jang [2],*

[1]  Biotechnology Research Division, National Institute of Fisheries Science, Busan 46083, Republic of Korea
[2]  Biopharmaceutical Engineering Major, Division of Applied Bioengineering, Dong-Eui University, Busan 47340, Republic of Korea
[3]  Department of Biotechnology, Pukyong National University, Busan 48513, Republic of Korea
*   Correspondence: wjjang@deu.ac.kr
†   These authors contributed equally to this work.

**Abstract:** The fish gut microbiota plays an important role in overall health. However, few reports have described the changes in the composition of gut microbiota following infection with pathogenic bacteria in olive flounder (*Paralichthys olivaceus*). Here, we reported the changes in the gut microbiota composition of flounder after treatment with each of the three pathogenic bacteria (*Edwardsiella tarda*, *Streptococcus iniae*, and *Vibrio harveyi*). *Edwardsiella tarda* infection decreased the relative abundance of Verrucomicrobia and increased Proteobacteria abundance at the phylum level of the gut microbiota over time. Similarly, *Streptococcus iniae* infection reduced the relative abundance of Verrucomicrobia. *Vibrio harveyi* infection caused a decrease in the relative abundance of Firmicutes and Verrucomicrobia and increased Proteobacteria. At the genus level, infection with all three pathogens increased the relative abundance of *Ralstonia* and *Sphingomonas* species. Conversely, this infection decreased the relative abundances of *Rubritalea*, *Saccharimonas*, and *Bacillus* species. Therefore, reducing the abundance of *Ralstonia* and *Sphingomonas* and increasing the abundance of *Rubritalea*, *Saccharimonas*, and *Bacillus* in the gut microbiota composition of flounder might help maintain a healthy gut microbiota balance. This research might be useful for future studies on improving the health of flounder through gut microbiota regulation.

**Keywords:** olive flounder; microbiome; *Edwardsiella tarda*; *Streptococcus iniae*; *Vibrio harveyi*

**Key Contribution:** Few reports have described changes in the composition of gut microbiota following infection with pathogenic bacteria in olive flounder. Here, we reported changes in the gut microbiota composition of olive flounder after infection with each of the three different pathogenic bacteria, and these findings might be useful for future studies on improving the health of flounder through gut microbiota regulation.

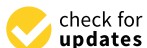



## 1. Introduction

Olive flounder (*Paralichthys olivaceus*) is an economically valuable fish species in the South Korean aquaculture industry; over the last five years, it has an average annual production of 41,480 tons, representing about 48.24% of the nation's total finfish production [1]. However, fish mortality caused by bacterial disease occurs every year, resulting in considerable economic losses to the aquaculture industry [2]. Representative bacterial diseases occurring in this farmed fish include edwardsiellosis, vibriosis, and streptococci [3]. Although antibiotics are used to treat these bacterial diseases, the ones used in flounder aquaculture cause the emergence of antibiotic-resistant pathogens and have residual effects on humans [4,5]. Therefore, microbial disease control strategies should be developed and

implemented in aquaculture to eliminate the use of currently used antibiotics, ensure food biosafety, and maintain human health [6].

The microbial community in the fish gut is altered by various factors, such as water quality, habitat, growth stage, bacterial and viral infections, and host feeding activity, which plays an important role in the overall health of fish [7]. The study and control of gut microbiome disruption are important factors in fish aquaculture [7,8]. A well-established and balanced gut microbiota provides essential fish developmental functions while protecting against pathogenic bacteria, which are among the leading causes of mortality in fish production systems [7,8]. However, the composition of fish gut microbiota is poorly explored and difficult to study because of large individual differences [9,10]. Therefore, in this study, the intestinal microbiota composition of olive flounder was investigated by pooling fish intestines together following controlled infection with pathogenic bacteria (*Edwardsiella tarda*, *Streptococcus iniae*, and *Vibrio harveyi*) that cause edwardsiellosis, streptococci, and vibriosis. The microbial composition and quantity were analyzed and compared between the different pathogenic bacterial infected groups.

## 2. Materials and Methods

### 2.1. Pathogenic Bacterial Strains and Culture Conditions

Three pathogenic bacteria (*Edwardsiella tarda* KCTC 12267$^T$, *Streptococcus iniae* KCTC 3657$^T$, and *Vibrio harveyi* KCTC 12724$^T$) were obtained from the Korean Collection for Type Cultures (KCTC, Daejeon, Republic of Korea) and cultured at 37 °C in brain heart infusion (BHI; Difco, Detroit, MI, USA) and marine broths (MB; Difco, Detroit, MI, USA), respectively. The stock cultures were stored at −70 °C in each broth containing 50% glycerol until use.

### 2.2. Animals, Sampling, and Infection Experiment

A total of 300 juvenile olive flounder with an average weight of 10.27 ± 0.22 g were purchased from a commercial flounder hatchery (Jeo-Gu fish farm, Geoje, Republic of Korea) and acclimatized in 120 L semi-recirculating tanks for 1 week. After acclimatization, 270 fish with no visible problems were equally distributed into 3 groups (30 fish/tank, triplicates). The intestines from the remaining 30 fish were collected for initial fish gut microbiota analysis. Then, 0.1 mL of the prepared pathogens diluted in PBS ($1.0 \times 10^8$ cfu/mL) was injected via intraperitoneal inoculation. The intestines of the infected fish were collected on days 2, 5, and 7 (10 fish/tank, triplicates). The water quality was regularly monitored twice a day, and stable environmental parameters were maintained: temperature, 19.0 ± 0.5 °C; pH, 7.2 ± 0.2; dissolved oxygen, 5.9 ± 0.4 mg/L; salinity, 30.5 ± 1.0 g/kg; and water flow, 1.2 L/min. Seawater was not shared between groups and was provided after pre-filtration and UV disinfection to prevent infection with other microorganisms.

### 2.3. Intestinal Microbiota Analysis

The total microbial DNA was isolated from the intestines of pathogen-infected olive flounder by using ZymoBIOMICS DNA Miniprep kits (Zymo Research Corp., Irvine, CA, USA). The quality of the total DNA was assessed through gel electrophoresis, and the V3-V4 region of 16S rRNA was amplified using primers containing the Illumina overhang adapter sequence to construct a library (forward primer: 5′–TCGTC GGCAG CGTCA GATGT GTATA AGAGA CAGCC TACGG GNGGC WGCAG–3′, reverse primer: 5′–GTCTC GTGGG CTCGG AGATG TGTAT AAGAG ACAGG ACTAC HVGGG TATCT AATCC–3′, Illumina, San Diego, CA, USA). The KAPA HiFi HotStart ReadyMix (KAPA Biosystems, Woburn, MA, USA) and Agencourt AMPure XP system (Beckman Coulter Genomics, USA) were used for PCR and purification of the PCR product, respectively. PCR was performed with a Veriti 96-well thermal cycler (Applied Biosystems, Foster City, CA, USA) at the Core Facility Center for Tissue Regeneration, Dong-Eui University (Busan, South Korea). The prepared library was quantified (Qubit dsDNA HS Assay Kit, Thermo Scientific, Waltham, MA, USA), quality controlled (Agilent 2100 bioanalyzer, Agilent Tech-

nologies, Waldbronn, Germany), and sequenced (Illumina MiSeq system, 300 bp paired end reads) at the Moagen (Daejeon, South Korea). The data were analyzed using the EzBioCloud server (http://www.ezbiocloud.net/, accessed on 4 January 2023). A heatmap analysis of genus abundance was performed using a heat mapper (http://www.heatmapper.ca/, accessed on 24 March 2023).

### 2.4. Statistical Analysis

The statistical significance of the data was analyzed by one-way analysis of variance (ANOVA) using Statistical Package for the Social Sciences (SPSS; IBM, Armonk, NY, USA), followed by Duncan's multiple range test. A *p*-value of < 0.05 was considered significant.

## 3. Results

### 3.1. Changes in Gut Microbiota Composition at the Phylum Level

Changes in the composition of the gut microbiota at the phylum level following pathogen infection were investigated according to the pathogen species and sampling time. The most abundant phylum in the composition of the gut microbiota of the initial group before pathogen infection was Firmicutes (71.42 ± 6.76%), followed by Verrucomicrobia (14.73 ± 4.88%), Proteobacteria (7.97 ± 1.69%), Saccaribacteria (2.59 ± 0.48%), and Bacteroidetes (1.81 ± 0.64%). *E. tarda* infection gradually decreased the abundance of Verrucomicrobia and increased the abundance of Proteobacteria over time. On day 7 of infection, the abundance of Proteobacteria increased to 23.37 ± 6.78% (Figure 1a). *S. iniae* infection also reduced the abundance of Verrucomicrobia, which was 6.38 ± 0.18%, 0.45 ± 0.04%, and 0.01 ± 0.01% on days 2, 5, and 7, respectively (Figure 1b). *V. harveyi* infection gradually decreased the abundance of Firmicutes and Verrucomicrobia but increased the abundance of Proteobacteria (Figure 1c).

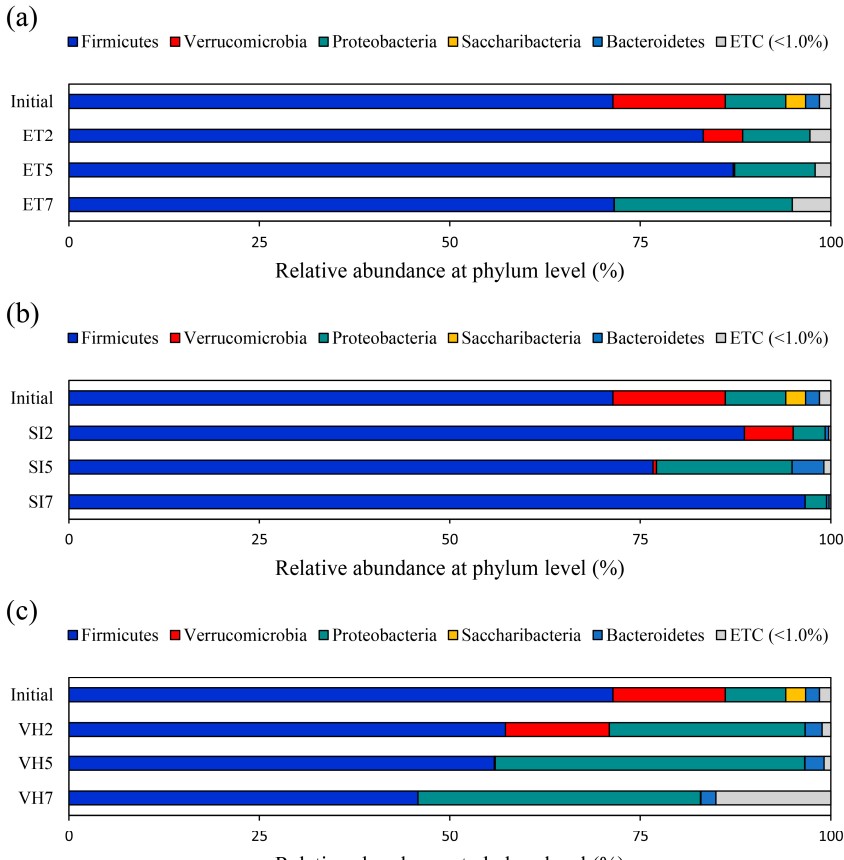

**Figure 1.** Composition and relative abundance of the intestinal bacterial communities of olive flounder according to infection with each of the three pathogens ((**a**) *Edwardsiella tarda*; (**b**) *Streptococcus*

*iniae*; (**c**) *Vibrio harveyi*) at the phylum level. Intestinal bacterial communities were analyzed via next-generation sequencing by isolating the total DNA of microorganisms present in the gut of olive flounder on days 0 (initial), 2, 5, and 7 after artificial infection with pathogens (*n* = 10, triplicates). ET, *Edwardsiella tarda*; SI, *Streptococcus iniae*; VH, *Vibrio harveyi*.

### 3.2. Changes in Gut Microbiota Composition at the Genus Level

Infection with *E. tarda* or *S. iniae* did not significantly (Duncan's multiple range test, *p* > 0.05) increase the proportion of *E. tarda* or *S. iniae* in the gut microbiome, respectively. Conversely, *V. harveyi* infection increased *Vibrio* species abundance over time (Figure 2). In all three groups, the infection commonly increased the proportions of *Ralstonia* and *Sphingomonas* but decreased the proportions of *Rubritalea*, *Saccharimonas*, and *Bacillus* (Figures 2 and 3).

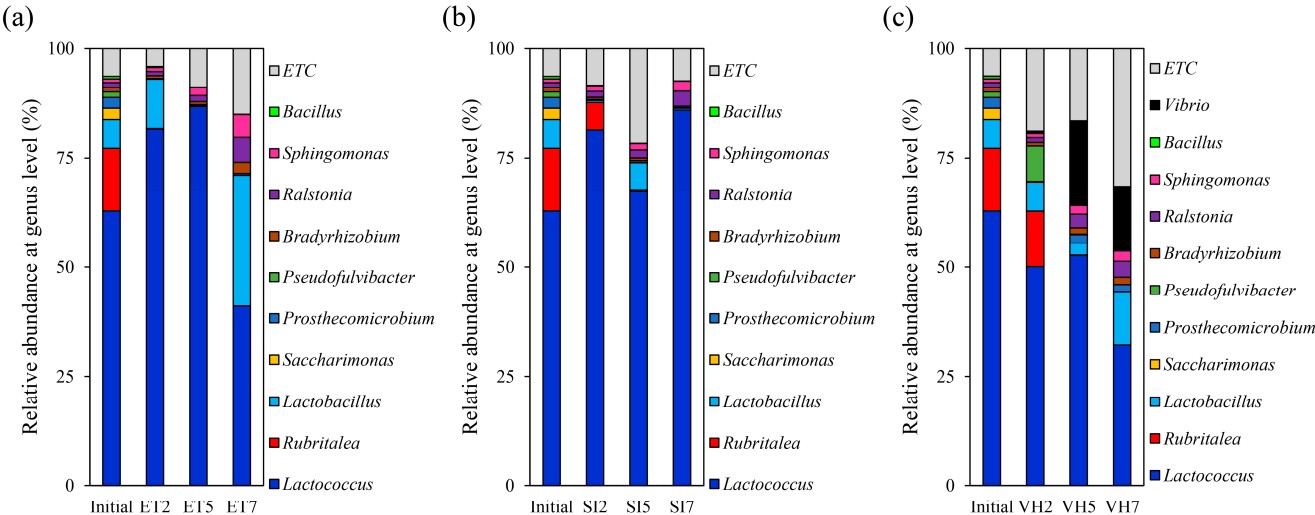

**Figure 2.** Composition and relative abundance of the intestinal bacterial communities of olive flounder according to infection with each of the three pathogens ((**a**) *Edwardsiella tarda*; (**b**) *Streptococcus iniae*; (**c**) *Vibrio harveyi*) at the genus level. Intestinal bacterial communities were analyzed via next-generation sequencing by isolating the total DNA of microorganisms present in the gut of olive flounder on days 0 (initial), 2, 5, and 7 after artificial infection with pathogens (*n* = 10, triplicates). ET, *Edwardsiella tarda*; SI, *Streptococcus iniae*; VH, *Vibrio harveyi*.

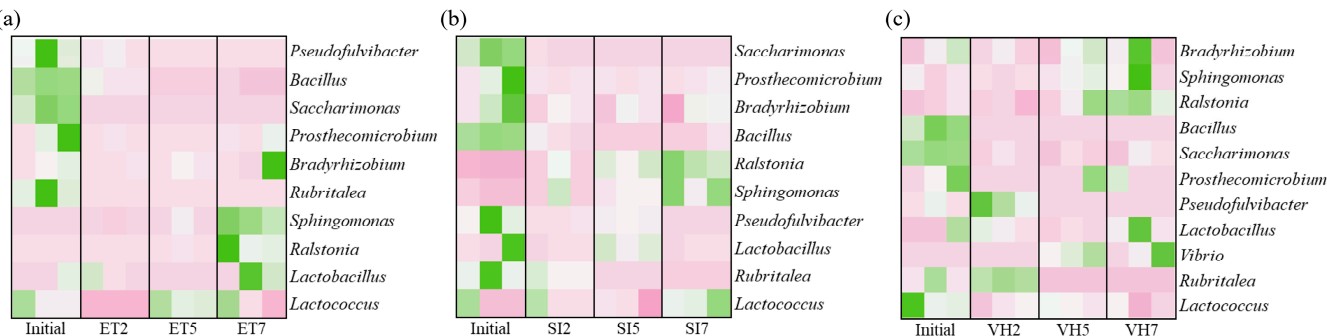

**Figure 3.** Heatmap analysis of the abundance of genera in the intestinal microbiota of olive flounder according to pathogen ((**a**) *Edwardsiella tarda*; (**b**) *Streptococcus iniae*; (**c**) *Vibrio harveyi*) sampling time. Green represents more abundant genera in the corresponding sample, and pink represents less abundant genera.

### 3.3. α-Diversity Changes with Pathogen Infection

α-Diversity changes according to pathogen infection were investigated according to pathogen species and sampling time. Until the 7th day after infection, *E. tarda* infections did not cause significant differences (Duncan's multiple range test, *p* > 0.05) in richness (ACE and Chao1) and diversity (Shannon and Simpson) estimates (Figure 4a). *S. iniae* infection repeatedly decreases and increases in all estimates over time (Figure 4b). *V. harveyi* infection caused a significant difference (*p* < 0.05) only in the richness estimate, which was generally increased (Figure 4c).

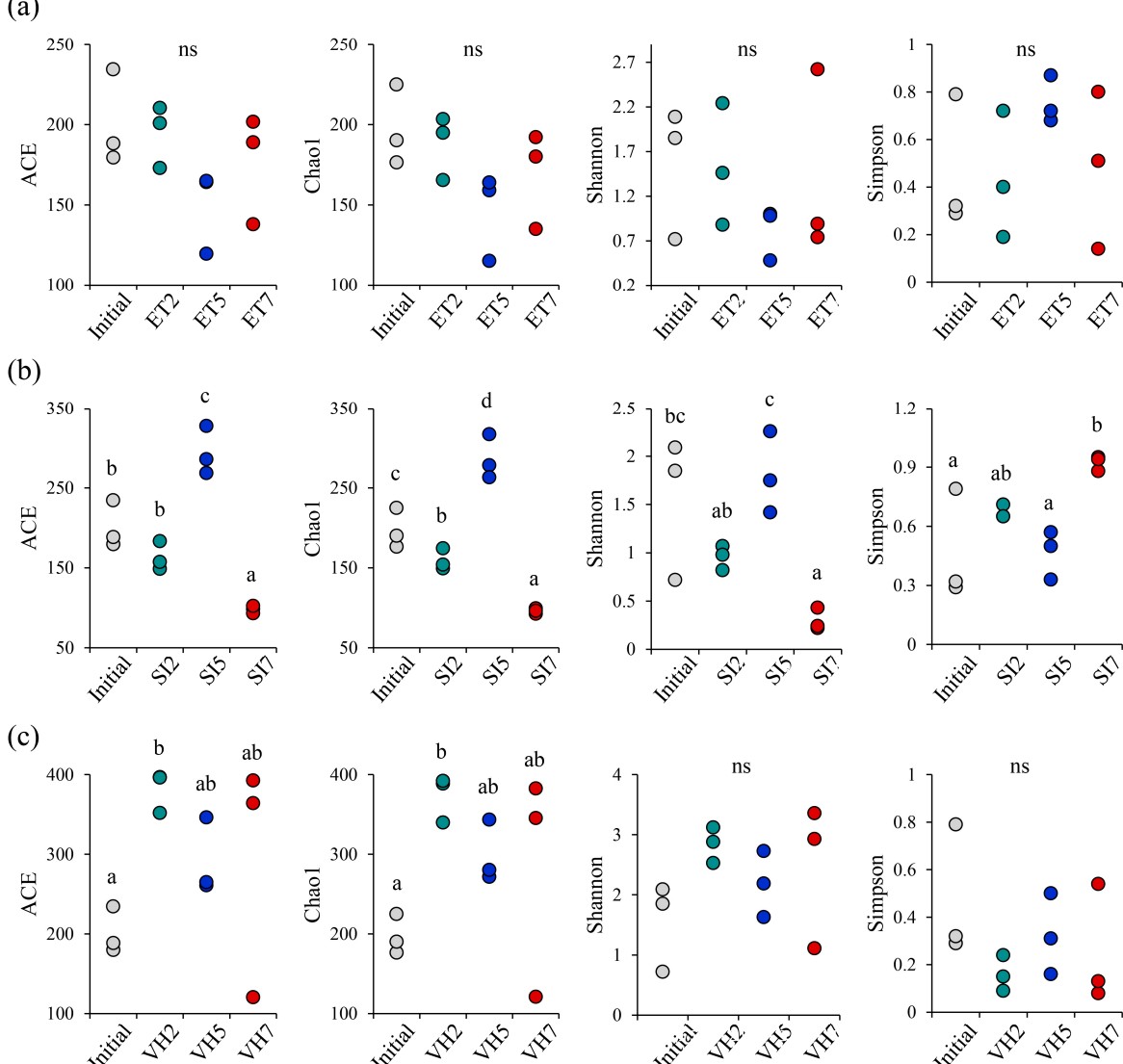

**Figure 4.** α-Diversity of the intestinal bacterial communities of olive flounder according to infection with each of the three pathogens ((**a**) *Edwardsiella tarda*; (**b**) *Streptococcus iniae*; (**c**) *Vibrio harveyi*). Changes in α-diversity at 0 (initial), 2, 5, and 7 days after infection with the pathogen were investigated in terms of richness and diversity estimates (*n* = 10, triplicates). Measurements with statistical differences are indicated by different letters, and measurements with no significant differences (*p* > 0.05) are indicated by ns. ET, *Edwardsiella tarda*; SI, *Streptococcus iniae*; VH, *Vibrio harveyi*.

## 4. Discussion

The gut of fish contains 21 bacterial phyla, with 3 dominant phyla (Proteobacteria, Firmicutes, and Cyanobacteria) accounting for over 70% [11]. These gut microbes provide their host with various beneficial effects, including pathways for energy harvesting, essential vitamin production, intestinal maturation, and immune system development [12–14].

When the host is healthy, the gut microbiome is stable, but diseases related to metabolism and immune response cause an imbalance of the microbiome [12–14]. Therefore, controlling the gut microbiome is important because maintaining its balance can protect the host from disease and provide essential developmental functions [8].

Many Gram-negative fish pathogenic bacteria, such as *E. tarda*, *E. ictaruli*, *V. harveyi*, *V. anguillarum*, *V. parahaemolyticus*, *V. ordalii*, *Aeromonas hydrophila*, *A. veronii*, and *Pseudomonas anguilliseptica*, have been reported to belong to Proteobacteria [15]. Zhao et al. (2022) showed that the abundance of Proteobacteria increases because of an energy imbalance in the intestinal microbial composition of the Songpu mirror carp (*Cyprinus carpio* L.) when subjected to starvation stress [16]. Similarly, Tran et al. (2018) reported that Proteobacteria are associated with unstable gut microbiota, and their increase is a potential diagnostic criterion for dysbiosis and disease [17,18]. In the present study, infection with each of the three pathogens commonly increased the abundance of Proteobacteria in the gut microbiota. In particular, infection with *V. harveyi* showed an increasing trend over time. Thus, starvation stress and pathogen infections cause unstable gut microbiota and lead to an increase in the abundance of Proteobacteria [16]. According to previous studies, members of Verrucomicrobia can process decaying organic matter and polysaccharides and participate in the digestion of plant cellulose in fish intestines [19,20]. The reduction of the abundance of these microorganisms following antibiotic treatment determines the decrease in the carp's cellulase activity [21], which can adversely affect digestion as a result. In this regard, in this study, pathogenic infection reduced Verrucomicrobia, leading to the reduced digestibility of fish, which could result in an imbalanced health status.

*Ralstonia* is a Gram-negative bacterial genus, and the most common member, *R. pickettii*, is an important human pathogen that causes infections such as osteomyelitis and meningitis and can pose a threat to seafood safety [22]. This genus has been found in fish such as sea bass (*Dicentrarchus labrax*) [23], yellow catfish (*Pelteobagrus fulvidraco*) [24], and rainbow trout (*Oncorhynchus mykiss*) [25], but its fish pathogenicity has not yet been reported. In a previous study, cases of infection with *Spingomonas echinoides* were identified in the intestines of moribund rainbow trout with typical symptoms of bacterial hemorrhagic sepsis [26]. Some *Spingomonas* species may be beneficial to fish, but other species may be pathogenic to other organisms [26]. In the present study, pathogen infection increased the abundance of *Ralstonia* and *Sphingomonas* in the gut microbial composition of olive flounder. Our results indicated that these genera might be associated with unhealthy conditions that disrupt the intestinal microbial balance; particularly, *Spingomonas* is the representative infection-associated genus. However, further research is needed to determine if these bacteria actually have detrimental effects on fish health.

The abundance of *Rubritalea*, *Saccharimonas*, and *Bacillus* species decreased because of pathogen infection. Among them, *Bacillus* is found in the intestines of healthy fish; because it exerts beneficial effects such as controlling the growth of opportunistic pathogens and producing antiviral compounds on hosts, many studies have been conducted on its biotechnological uses, such as probiotics [12]. Although few studies have been performed on the relationship between fish and other microorganisms (*Rubritalea* and *Saccharimonas*) that decrease with pathogen infection, at least *Bacillus* appears to be an important strain that must be present to maintain the balance of intestinal microbiota in fish [12].

## 5. Conclusions

In conclusion, changes in the intestinal microbiota composition of olive flounder were observed through controlled infection with each of the three pathogens. At the phylum level, Proteobacteria increased, and Verrucomicrobia decreased. At the genus level, *Ralstonia* and *Sphingomonas* increased, and *Rubritalea*, *Saccharimonas,* and *Bacillus* decreased. This result is attributed to an imbalance of intestinal microbes caused by infection with pathogens. However, additional studies are needed to determine whether the change in microbial abundance shown in this study actually has a significant effect on fish immunity and viability, and what mechanism works.

**Author Contributions:** Conceptualization, D.-G.K., S.-J.L. and W.J.J.; methodology, D.-G.K. and S.-J.L.; software, W.J.J.; validation, D.-G.K., S.-J.L. and W.J.J.; formal analysis, D.-G.K. and S.-J.L.; investigation, D.-G.K. and S.-J.L.; writing—original draft preparation, D.-G.K. and W.J.J.; writing—review and editing, J.M.L. and E.-W.L.; supervision, W.J.J.; project administration, D.-G.K. and W.J.J.; funding acquisition, D.-G.K. and W.J.J. All authors have read and agreed to the published version of the manuscript.

**Funding:** This work was supported in part by Basic Science Research Program through the National Research Foundation of Korea (NRF) funded by the Ministry of Education (NRF-2021R1I1A1A01049238), and in part by the National Institute of Fisheries Science, Ministry of Oceans and Fisheries, Korea (R2023019).

**Institutional Review Board Statement:** This study was conducted under the guidelines of the Animal Ethics Committee Regulation issued by Dong-Eui University. Approval code: (DEU-R2022-031).

**Data Availability Statement:** The datasets generated and/or analyzed in the current study are available from the corresponding author upon reasonable request.

**Conflicts of Interest:** The authors declare no conflict of interest.

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
