# Peer review of "Changes in the Gut Microbiota Composition of Juvenile Olive Flounder (Paralichthys olivaceus) Caused by Pathogenic Bacterial Infection"

_fishes, doi:10.3390/fishes8060294_

Round 1

Reviewer 1 Report

This is an interesting study on the changes in the composition of the fish gut microbiota as a response to challenge with different fish pathogens. The authors show that the microbiota changed differently in fish challenged with different pathogens.

I have some minor points, which I hope the authors might consider. All Figures (1-4) are divided into a, b and c. It took a while until I was able to find what a, b and c meant, as no explanation was given in the figure text. I would suggest that this explanation is added in the figure text. It is not sufficient that the groups are numbered according to the pathogens.

Author Response

We have received your considerate review of the manuscript. It was a great pleasure to see your insightful and helpful comments. We hope our responds to your comments finds well in addressing most of the mentioned inquiries.  Please check the attached file for responses and corrections to comments.

Reviewer 2 Report

The manuscript "Changes in the Gut Microbiota Composition of Juvenile Olive Flounder (Paralichthys Olivaceus) Caused by Pathogenic Bacterial Infection" (Manuscript ID: fishes-2355030) presented the results of a 16S metabarcoding study on the intestinal microbiome of juvenile olive flounders following experimental infection with three different pathogenic bacteria (E. tarda, S. iniae, V. harveyi). 

The manuscript presents various gaps and aspects that must be implemented in order to be taken into consideration for possible publication.

Major comments

- The introduction does not present a description of the pathogenic bacteria with which the infection is carried out and a reason regarding the choice of these pathogens.

- There is no mention of a negative control (uninoculated fishes) in the materials and methods, which therefore does not allow to correctly evaluate the results of experimental infections. In fact, it cannot be excluded that the conditions of artificial housing cause stress for the fish, with consequent variation of the microbiota: for this reason, a negative control is necessary. 

- The description of the methods is low detailed: mention the reagents used for the amplification (Taq, master mix), for the index PCR, for the quality controls. Was the sequencing done single-end or paired-end?

- Statistical analysis is mentioned in the results, but nothing is said in the materials and methods part: this section needs to be implemented.

- Are the relative abundances reported in the results purely descriptive or statistically significant? Specify this aspect.  

Minor comments

- Line 14: add the scientific name of olive flounder.

- Line 15: specify the pathogenic bacteria used in the experimental infections, as they are not previously mentioned.

- Keywords: eliminate "pathogen" and one between "microbiome" and "microbiota". Enter the names of the pathogens. 

- Line 43: delete "at a field level".

- Line 44: delete "synthetic".

- Lines 48-52: rephrase the sentence. 

- Line 109: what is meant by "significantly" if the result is not supported by statistics?

- Lines 139-141: delete the sentence. 

- Line 150: "Many Gram-negative fish pathogenic bacteria have been reported to belong to Proteobacteria": provide examples. 

- Line 169: add the scientific names of the indicated fishes. 

Moderate editing of English language 

Author Response

We greatly appreciate to the Reviewer for evaluating our manuscript. We tried to respond to the honorable reviewer comments point-by-point, indicating necessary changes in the revised version. We believe these changes have substantially improved the quality and clarity of this article. Please check the attached file for responses and corrections to comments.

Reviewer 3 Report

Review

 Changes in the Gut Microbiota Composition of Juvenile Olive 2 Flounder (Paralichthys Olivaceus) Caused by Pathogenic 3 Bacterial Infection

This research paper is in the journal scope. The authors did interesting work in regards to the microbial gut communities in bacterial infection. That said, much information is missing from the paper.

The main problems also described in some details below, are the missing information regarding control groups and initial sample time?  Was there a difference between the sampling points in the clinical conditions of the fish after infection? Were there any apparent pathologies seen? Were there samples isolated from infected fish to examine the presents of the original bacteria which they were infected by? The triplicates are not clear were intestines from all triplicates from the same group polled together or just the fish samples from the same triplicate? In the results it says that “not significant different” were see; but there is no description in the text or figures of the statistical program that was used? A good description of what was estimated by ACE, Chaol, Shannon, Simpson should be included in the M&M as well. Part of the discussion and the conclusion should relate to what was actually described in the study and not on assumption.

 More specific suggestions and remarks:

 Key Contribution: Few reports have described changes in the composition of gut microbiota following infection with pathogenic bacteria in olive flounder. Here, we reported changes in the gut microbiota composition of olive flounder after infection with three different pathogenic bacteria. These findings might be useful for future studies on improving the health of flounder through gut microbiota regulation.

Introduction

Line 52:  Suggested change: However, the composition of fish gut microbiota is poorly explored and difficult to study because of large individual differences. Add a reference to that statement

Lines 54-56: Suggested change:  Therefore, in this study, the intestinal microbiota composition of olive flounder was investigated by pooling fish intestines together following controlled infection with pathogenic bacteria. The microbial composition and quantity was analyzed and compered between the different pathogenic bacterial infected groups.

Material and Methods

Line 62: brain heart infusion, marine broths,- Add company, producer and or manufacture name and country.

Line 69: healthy fish- How was it established that these fish were healthy? What analysis was done? need to specify.

Line 69-70: Suggested change: Then, 0.1 mL of the prepared pathogens (1.0 × 108 cfu/mL) was injected via intraperitoneal inoculation. What was in the injected preparation with the bacteria (pbs? Media?)?

Line 71: The intestines of the infected fish were collected on days 2, 5, and 7 (10 fish/tank, triplicates). What about time 0, before infection, as a control to have comparison? As described in the introduction the fish intestine were pulled together, weren’t they? Need to be add it in the M&M as well.

Line 80: V3-V4 region- of what gene? I presume 16S but that need to be stated.

Main missing information:

Was there a control group? Was there any initial analysis before infection (it sates so in the result but it is not described in the M&M)?

Result:

Line 91: infection time. -  Isn’t it more accurate to say sampling time? Not infection time.

Line 109: significantly increase- what analysis was mad to establish significant? it should also be described in the M&Ms. It is only mentioned in 3.3.α-. Diversity changes with pathogen infection.

Figure 1 and 2. Add a,b and c according to the bacteria presented in the figure in the description of the legends (legends of figures should stand alone)

Also add the type (name) for the Composition and relative abundance.

Add n=? and number and triplicates, to figures

Figure 3. Infection time should change to sampling points. a,b and c should be described (legends of figures should stand alone).

Figure 4. α-Diversity of the intestinal bacterial communities of olive flounder according to infection 133 with each of the three pathogens. Changes in α -diversity at 0 (initial), 2, 5, and 7 days after infection 134 with the pathogen were investigated in terms of richness and diversity estimates. Measurements 135 with statistical differences are indicated by different letters, and measurements with no significant 136 differences are indicated by ns. ET, Edwardsiella tarda; SI, Streptococcus iniae; VH, Vibrio harveyi.

Add a,b and c according to the bacteria presented in the figure in the description of the legends (legends of figures should stand alone)

Also add the type (name) for the Composition and relative abundance.

Add n=? and number and triplicates, to figures

Also add description of the letters presenting significant differences as well as what is ns?

Shannon is spelled incorrectly on the graphs.

Discussion:

Line 158: Thus, starvation stress and pathogen infections cause unstable gut microbiota and lead to an increase in the abundance of Proteobacteria- Add reference

Line 164- In this study, pathogenic infection reduced Verrucomicrobia, leading to the reduced digestibility of fish, which could result in an imbalanced health status.-  This is not clear?  The author did not describe how digestibility in fish was analyzed but they did say that they could see Verrucomicrobia leading to reduced digestibility in this study. Need to clarify this statement.

Lines 175- 176: Our results indicated that these genera might be associated with unhealthy conditions that disrupt the intestinal microbial balance; particularly, Spingomonas is the representative infection-associated genus.

I would be carful indicating that, no clinical signs or fish health condition was described in the results in this paper and this increase in abundance of Ralstonia and Sphingomonas  can also be contributed to microbial coemption if the fish didn’t display any clinical signs and if the author had not isolated a pathogenic strain  from these genus, I would not say that these results indicate association to unhealthy condition of fish.

Lines 182-184: Although few studies have been performed on the relationship between fish and other microorganisms (Rubritalea and Saccharimonas) that decrease with pathogen infection, at least Bacillus appears to be an important strain that must be present to maintain the balance of intestinal microbiota in fish.- Add references

Conclusions

Line 188: Suggested change: artificial infection- Controlled infection

Lines 192-193:  if the balance of microbes in the gut of fish is maintained by controlling the increased or decreased microbes, then the fish’s immunity can be restored, and its health can be improved.

I do not see support of immunological assays done or clinical signs if infected fish described to give that conclusion. There can be many other reasons for the changes in gut microbiome.  

Moderate editing of English language

Author Response

(The authors gave the same response as above.)

Reviewer 4 Report

Dear authors,

Congratulations on this interesting manuscript, which brings a refreshing perspective on the gut microbiome change when challenged by a bacterial pathogen.

Overall the manuscript is well written. I only have some minor corrections that you can check on the attached document. I especially stress the Keyword selection part, as it is an important issue for the manuscript.

However, you must improve the Material and Methods section. It lacks a description of how you preserved your samples after sampling, and how you pooled your samples for analysis (as mentioned in the Introduction part corresponding to the objectives of this manuscript). This is a critical point to allow the reader to understand your experimental design. It lacks also a section describing how you performed the statistical analysis of the data, which is critical since you presented statistical analysis data in the Results section.

I also have two more questions for the authors regarding the experimental design:

- Why did you use an intraperitoneal injection of the bacteria and not a bath as a pathogen inoculation methodology in this experiment?

- Do you think introducing a Control treatment, inoculated with sterile saline solution, could be important to rule out gut microbiome changes due to confinement stress and inoculation methodology?

Best regards,

Dear authors,

Regarding the quality of the English language, overall the manuscript is well written. I have only some suggestions regarding some of the sentences, that are somewhat confusing or do not transmit a clear idea to the reader.

Please check them on the attached document also.

Best regards,

Author Response

Dear reviewer 4

We greatly appreciate to the Reviewer for evaluating our manuscript. We tried to respond to the honorable reviewer comments point-by-point, indicating necessary changes in the revised version. We believe these changes have substantially improved the quality and clarity of this article. The manuscript has been revised in its entirety by referring to the attached file. Please check the attached file for responses and corrections to comments.

- Why did you use an intraperitoneal injection of the bacteria and not a bath as a pathogen inoculation methodology in this experiment?

Response: We aimed to observe rapid changes in gut microbiota according to pathogenic strains. However, like your suggestion, I think that a study on the effect of immersion in pathogens is also very valuable. If possible, we will also design a study that directly compares the two methods.

- Do you think introducing a Control treatment, inoculated with sterile saline solution, could be important to rule out gut microbiome changes due to confinement stress and inoculation methodology?

Response: We totally agree with you. If the gut microbiome of fish injected intraperitoneally with PBS were sampled on days 2, 5, and 7, many results would have been obtained. However, in this study, only differences in gut microbiota according to differences in pathogenic strains were compared.

We thank you again for your consideration of this manuscript. Please do not hesitate to contact me directly if necessary.

Yours sincerely,

Authors,

Round 2

Author Response

Thank you very much for your kind review. The answer can be found in the attached file below, and the modified part has been marked in red in the revised manuscript.

Reviewer 4 Report

Dear authors.

First of all, thank you for considering all my suggestions and for the answers to my question in the previous review. I really think that you should proceed with a comparison between both methods of inoculation, which can give you very valuable data for a possible new manuscript.

And let me congratulate you on the great improvement of the manuscript. Is way more clear and readable. I only have some minor corrections regarding the species name format, which you can check in the attached document.

Best regards,

Dear authors,

Regarding your English writing, it has improved a lot. I only have some minor corrections, mostly regarding the species name format (please check the attached document).

Best regards,

Author Response

We are grateful that our manuscript is being improved thanks to your insightful and useful comments. Corrected parts are marked in red in the revised manuscript.

(The bacterial notation has not been changed, and will be changed later according to the journal's request by referring to the journal regulations.)

Thank you very much for your kind review. Please do not hesitate to contact me directly if necessary.

Yours sincerely,

Authors,